# Uterus Preservation in Case of Vaginal Prolapse Surgery Acts as a Protector against Postoperative Urinary Retention

**DOI:** 10.3390/jcm9113773

**Published:** 2020-11-23

**Authors:** Christine Bekos, Raffaela Morgenbesser, Heinz Kölbl, Heinrich Husslein, Wolfgang Umek, Klaus Bodner, Barbara Bodner-Adler

**Affiliations:** 1Division of General Gynecology and Gynecologic Oncology, Medical University of Vienna, Waehringer Guertel 18-20, A-1090 Vienna, Austria; christine.bekos@meduniwien.ac.at (C.B.); raffaela.morgenbesser@meduniwien.ac.at (R.M.); heinz.koelbl@meduniwien.ac.at (H.K.); heinrich.husslein@meduniwien.ac.at (H.H.); Wolfgang.Umek@meduniwien.ac.at (W.U.); klausbodner@yahoo.com (K.B.); 2Karl Landsteiner Society for Special Gynaecology and Obstetrics, Medical University of Vienna, Waehringer Guertel 18-20, A-1090 Vienna, Austria

**Keywords:** pelvic organ prolapse, post-void residual, postoperative urinary retention, prolapse hysterectomy, uterus preserving prolapse surgery

## Abstract

Background: The aim of this study was to identify clinical risk factors for increased post-void residual (PVR) volumes in patients undergoing vaginal prolapse surgery and to find out whether uterus preservation or prolapse hysterectomy influences the incidence of postoperative urinary retention. Methods: This retrospective study included women who presented with pelvic organ prolapse (POP) and planned prolapse surgery between January 2017 and July 2019. PVR was assessed postoperatively and increased amounts were defined as incomplete voiding with residual urine volume greater than 150 mL. Results: Increased PVR at the first postoperative day occurred in 31.8% (56/176). Body mass index (BMI) was significantly lower in patients with increased PVR after pelvic floor surgery compared to patients with normal PVR amounts (*p* = 0.040). Furthermore, during multiple logistic regression analysis, low BMI (*p* = 0.009) as well as prolapse hysterectomy (*p* = 0.032) turned out to be the strongest risk factors associated with increased PVR volume. Conclusion: This is the first study identifying prolapse hysterectomy as an independent risk factor for increased PVR after surgical prolapse repair. Our results might be helpful in counseling patients prior to surgery and underline the option of uterus preservation during prolapse surgery in selected cases.

## 1. Introduction

Pelvic organ prolapse (POP) is a common condition that increases with age and affects every second elderly woman [1]. In general, treatment options for patients with symptomatic POP include, beside expectant management, primarily pessary placement and surgical repair [2]. On the one hand, prolapse surgery reduces pelvic floor symptoms by restoring the anatomy of the vagina and the surrounding visceral organs, on the other hand, surgical complications, concomitants and side effects can occur [3]. The current life-time risk of undergoing reconstructive surgery for pelvic organ prolapse is reported to be 19% for the female population [4].

It is well known that after pelvic floor surgery, an elevated risk for voiding dysfunction or postoperative urinary retention (POUR) exists. Incidence of urinary retention following any type of vaginal prolapse surgery varies between 6% and 29% in literature [5,6,7]. Tissue swelling, inflammation, as well as damage to peripheral nerve endings during pelvic floor surgery might be responsible for transient postoperative voiding dysfunction (PVD) with increased post-void residual (PVR) volume [8].

So far, no standardized PVR definition exists. However, consensus exists that a PVR amount of 50 mL to 100 mL is defined as normal, whereas PVR amounts greater than 200 mL are specified as significant or abnormal and require intervention. Furthermore, no standardized treatment strategies in case of increased post-void residual urine volume exist, although postoperative catheterization for acute urinary retention represents a significant burden for affected patients [9], increases the risk of urinary tract infection, and delays the recovery process with prolonged hospital stay. So far, known risk factors for POUR include old age [10], high grade cystocele, severe intraoperative blood loss, postoperative pelvic hematoma, as well as low BMI [6,11,12]. In addition, the effects of general anesthesia on detrusor activity [13], perioperatively used medications such as opioids [14], and the operation itself with manipulation of tissue and nerves near the genitourinary system are defined as further risk factors [15]. Over the past years, percentage of uterine-preserving prolapse surgeries has increased as this surgical method was shown to be non-inferior compared to prolapse hysterectomy [16]. Due to further advantages like less anatomical recurrences and repeat surgeries [17], uterus preservation is gaining popularity and the number of transvaginal sacrospinous hysteropexy (SSH) is increasing, also at our institution.

The aim of the present study was to evaluate clinical risk factors for increased PVR amounts in patients with symptomatic POP undergoing vaginal prolapse surgery. Furthermore, the authors focused on the question whether uterus preservation (with SSH) or hysterectomy during prolapse surgery influences the incidence of postoperative urinary retention with significant PVR amounts.

## 2. Material and Methods

This retrospective study included all women who presented with POP and planned prolapse surgery at the Department of General Gynecology and Gynecologic Oncology, Medical University of Vienna, with recruitment between January 2017 and July 2019.

The study was approved by the Ethics Committee of the Medical University of Vienna (IRB nb: 2080/2019) before the study was initiated. Due to its retrospective design, the ethics committee waived the requirement to obtain distinct informed consent from patients.

### 2.1. Patients Characteristics

Eligible cases were women who visited our tertiary urogynecology unit for dominant prolapse symptoms and indication as well as wish for pelvic floor surgery. Correction of anterior and/or apical compartment during prolapse surgery was obligatory. Exclusion criteria were inadequate documentation and sole correction of the posterior compartment. Furthermore, patients with a history of pelvic floor surgery were excluded from the analysis.

The following characteristics were assessed for each patient: PVR amount on the first day after operation (twice per day) with bladder scan. In case of significantly increased amounts of PVR, frequency of clean intermittent catheterization was documented, as well as the duration until normalization of PVR. Furthermore, general characteristics like patient’s age, parity, body mass index (BMI), Pelvic Organ Prolapse Quantification (POP-Q) stage, the presence of comorbidities, American Society of Anesthesiologists (ASA) classification, diabetes mellitus, smoking status, type of anesthesia, type of prolapse surgery, operation time, blood loss during operation, and occurrence of postoperative complications were assessed. Postoperative complications were graded according to the Clavien–Dindo classification [18]. BMI was calculated by the formula: weight (kg)/height^2^ (m^2^). The extent of prolapse was assessed with an urogynecologic examination and documented according to the ICS POP-Q–system [19]. Preoperative PVR volume was obtained from all patients by straight catheterization within 10 min after spontaneous voiding.

### 2.2. Type of Prolapse Surgery

Surgical intervention of prolapse surgery consisted of correction of all the affected compartments and the decision which technique was used was left to the discretion of the responsible urogynecologist. As Austria has a strong tradition and experience in vaginal surgery, the planned surgery could include either vaginal prolapse hysterectomy with colporrhaphy and McCall Culdoplasty, or in case of uterus preservation, vaginal sacrospinous hysteropexy. All procedures were performed as native tissue repairs. At our institution, surgery is performed by 3 senior surgeons who are qualified and trained in these procedures. All operations were performed under general or spinal anesthesia and prophylactic antibiotics were given preoperatively.

### 2.3. Postoperative Management und Management of POUR

At the end of the operation, each patient received a sterile transurethral indwelling Foley catheter (CH 14) which was removed after 24 h as part of routine postoperative care. At our institution, a spontaneous voiding trial two times/day is performed on the first postoperative day. Each patient is instructed to wait to void until she has a strong urge to void, feels bladder fullness, or four hours have passed. After spontaneous voiding, PVR is measured by a bladder scan within 15 min of the completed void. In general, PVR is defined as the volume of urine left in the bladder after micturition [20]. At our institution, increased PVR is defined as failure of first voiding trial necessitating catheterization or presence of incomplete voiding. The cut-off value to perform clean intermittent catherization is a residual urine volume greater than 150 mL. Patients with elevated post-void residual volumes continue mechanical bladder drainage via intermittent clean self-catheterization until the post-void residual volumes are consistently (at least 2×/24 h) less than 100 mL. As no unique consensus of critical thresholds exist, management and cut-off values were decided in-house.

### 2.4. Statistics

Values are presented as mean values with standard deviation (SD) or as median with range. In order to compare presence and absence of PVR with clinical parameters, Student’s *t*-tests were performed. *p*-values of <0.05 were considered statistically significant. In this normally distributed population, a mean age of 62.4 was used as cut-off for further analyses. Multiple regression analysis was performed for identifying independent risk factors for POUR with PVR. The odds ratio (OR) estimates and their 95% confidence interval (CI) were calculated from the multivariable logistic regression model with forward conditional selection method. Statistical analyses were performed using the statistical software SPSS 25.0.

## 3. Results

A total of 176 documented cases with pelvic organ prolapse surgery were included. Mean age of all patients was 62.42 ± 12.06 years (range 28.0–96.0 years) and median body mass index was 26.50 ± 4.32 kg/m^2^. In 10/160 (6%) women, preoperative elevated PVR amounts were documented (mean 227 mL; range 110–290 mL), whereat 2 of these 10 women also showed increased volumes postoperatively.

Furthermore, 115/176 (65.3%) women underwent prolapse hysterectomy and 61/176 (34.7%) patients received uterus preserving prolapse surgery with vaginal SSH. In all, 18 events were defined as postoperative complications (10 cases with UTI or bacteriuria; 3 patients developed postoperative fever >38° within the first 72 h after surgery, 4 women suffered from 4 hematoma/abscess of the vaginal vault, and in one case bladder injury was performed).

Baseline characteristics of the study population are shown in Table 1.

### 3.1. Cases with POUR and Increased PVR Amounts

Increased post-void residual urine volume (PVR) at the first postoperative day occurred in 31.8% (56/176) of patients. Mean amounts of PVR at the first postoperative day were 277.37 mL. Moreover, 60.7% (34/56) of the patients with significant PVR needed intermittent clean self-catheterization. Time to normalization was 2.48 days.

Mean age of affected women with significant postoperative PVR was 63.93 ± 10.17 years (range: 33.0–85.0) and median BMI was 26.10 ± 3.74 (range 18.9–34.9).

Based on the POP-Q system [12], 18/56 (32.1%) cases were diagnosed with stage II prolapse, 8/56 (14.3%) were classified as stage III, and 30/56 (53.6%) as stage IV prolapse cases.

### 3.2. Comparison between Patients with Increased and Normal PVR after Surgical Prolapse Repair

BMI was significantly lower in patients with increased PVR after pelvic floor surgery compared to patients with normal PVR amounts (*p* = 0.040). Furthermore, age, type of prolapse surgery, POP-Q stage, ASA classification, anterior colporrhaphy, hypertension, diabetes, and postoperative complications did not significantly differ between patients with and without PVR amounts (*p* > 0.05) (Table 2).

Regarding postoperative complications, we observed mostly mild (Clavien Dino type I and II complications) postoperative complications in 18/176 patients. In 4/56 women with increased PVR amounts, we found mild postoperative complications. All four patients had postoperative urinary tract infection (UTI).

In the group of patients with PVR < 150 mL, we observed six women with UTIs/bacteriuria (presence of bacteria in urine), three cases showed postoperative fever >38° within the first 72 h after surgery, and five women suffered from hematoma/abscess of the vaginal vault. None of these patients underwent surgical revision, and in two cases, a re-admission was documented after seven days and all cases could be managed conservatively with analgetics and/or antibiotic parenteral treatment.

### 3.3. Multiple Logistic Regression Analysis

Multiple logistic regression analysis was conducted in order to define independent risk factors for significant PVR after surgical prolapse repair. The strongest risk factors associated with increased PVR volume were a low BMI (*p* = 0.009) as well as prolapse hysterectomy (*p* = 0.032) (Table 3).

### 3.4. Discussion

As a known elevated risk for voiding dysfunction or postoperative urinary retention after pelvic floor surgeries exists, the aim of the present study was to analyze risk factors for increased PVR in POP patients after their prolapse surgery. Furthermore, the authors focused on the research question whether the type of prolapse surgery (uterus preservation versus prolapse hysterectomy) influences the incidence of postoperative urinary retention with resulting significant PVR.

## 4. Main Findings

Our findings revealed increased PVR amounts in 31.8% of patients which is in line with data in incidence rates of postoperative urinary retention following any type of vaginal prolapse surgery of 6–29% [5,6,7]. Furthermore, the strongest risk factors associated with increased PVR volume were low BMI (*p* = 0.009) as well as prolapse hysterectomy (*p* = 0.032) in our cohort of affected patients. To the best of our knowledge, this is the first study identifying prolapse hysterectomy as an independent risk factor for increased PVR after surgical prolapse repair.

## 5. Comparison with Literature

In a retrospective analysis evaluating 332 patients undergoing reconstructive surgery for pelvic organ prolapse and/or stress urinary incontinence, next to other factors, anterior vaginal repair, laparoscopic colposacropexy, and vaginal hysterectomy were identified as risk factors for POUR [21]. Further, endoscopic hysterectomy was identified to double the risk for POUR compared to endoscopic non hysterectomy gynecologic surgery controls [22]. The route of hysterectomy, comparing laparoscopic, vaginal, and robotically assisted laparoscopic approaches, had no impact on the risk for urinary retention [23]. One may hypothesize that the operation itself with surgical tissue damage might be responsible for transient postoperative voiding dysfunction. Furthermore, we assume, as prolapse hysterectomy tends to result in more tissue swelling and inflammation compared to sacrospinous hysteropexy. This might be an explanation for the reduced risk of POUR in uterus preserving operation techniques.

Furthermore, we identified lower BMI as an independent risk factor for increased postoperative PVR amounts. This is supported by previous data showing that lower BMI was related to urinary retention in patients receiving tension-free vaginal tape [24].

Moreover, previous studies indicated that advanced prolapse may cause anatomic distortion of lower urinary tract including urethral kinking, resulting in elevated PVR amounts [25]. Tan et al. reported, for example, urinary splinting by 5–12% of women with stage II anterior prolapse and 23–36% of those with stage III/IV anterior prolapse [26]. Fitzgerald et al. also demonstrated in a retrospective analysis with 35 cases that preoperative voiding study had a sensitivity of 66%, a specificity of 46%, a positive predictive value of 12%, and a negative predictive value of 93% as a predictor of elevated postoperative PVR volumes [27]. In our study, increased preoperative PVR volume (>100 mL) was documented in 6% of included women, whereat two of these cases also showed elevated PVR amounts postoperatively. On the basis of our data and due to the different surgical technique groups (uterus preserving prolapse surgery and prolapse hysterectomy), the authors cannot draw serious conclusions regarding prediction of postoperative PVR after assessment of preoperative PVR amounts.

According to literature [5,6,7], our findings revealed increased PVR amounts in 31.8% of patients following vaginal prolapse surgery. In general, several trials regarding POUR after pelvic floor surgery exist [6,28,29], but the major difficulty in interpreting the literature is the lack of standardized definitions for “voiding dysfunction” and “urinary retention.” Additionally, the management of postoperative urinary retention is highly variable in the gynecologic community. There is no standardized and specific guideline regarding the appropriate time for urinary catheter removal after vaginal prolapse surgery. Alonzo-Sosa et al. reported no significant difference in the incidence of POUR in patients of which the urinary catheter was removed at the first postoperative day compared to the third day after vaginal prolapse surgery [30]. However, a randomized study evaluating patients following vaginal prolapse surgery demonstrated that patients after first-day catheter removal had a significantly higher incidence of postoperative urinary retention than those whose catheter was removed at the fifth day [31]. Although re-catheterization is significantly more frequent in one-day catheter-use, the incidence of urinary tract infection and longer duration of hospital stay increases with prolonged catheter indwelling time [32]. Therefore, early catheter removal for preventing urinary tract infection has been recommended. Similarly, the policy at our institution includes the early catheter removal after 24 h and a spontaneous voiding trial two times/day afterwards.

## 6. Limitations and Strengths of the Study

We are aware of the limitations of our study. Since this is a retrospective trial, selection bias might play a role. Thus, the authors can only comment on associations between elevated PVR volumes and risk factors but not on the underlying causality. As the management of POUR is highly variable in the urogynecologic community, a general limitation is resulting from different definitions as well as postoperative management practices performed at our institution and compared with practices at other institutions.

Although future prospective research is needed, this clinical study enlarges significantly the literature regarding the surgical prolapse method and influence on the occurrence of POUR.

## 7. Conclusions

Our study identified low BMI as well as prolapse hysterectomy as independent risk factors for increased PVR amounts in patients after vaginal prolapse surgery. Due to these findings, we assume that uterus preserving prolapse surgery acts as a protector against POUR. In our opinion, these findings could be another argument for uterus sparing surgery in selected prolapse cases and might be helpful in counseling the patients prior to surgery. Nevertheless, further studies are warranted to confirm these results in future prospective studies.

## Figures and Tables

**Table 1 jcm-09-03773-t001:** Patients’ characteristics of 176 included patients.

Age (years), mean (range)	62.42 (28.0–96.0)
BMI (kg/m^2^), median (range)	26.50 (17.8–39.0)
Parity, median (range)	2.0 (0.0–9.0)
Duration of operation (min), mean (range)	91.42 (20.0–390.0)
Type of anesthesia	
Spinal anesthesia, number (%)	2 (1%)
General anesthesia, number (%)	174 (99.0%)
Blood loss during operation	
<500 mL, number (%)	176 (100%)
>500 mL, number (%)	0 (0%)
Type of pelvic floor surgery	
Uterus preserving prolapse surgery, number (%)	61 (34.7%)
Prolapse hysterectomy, number (%)	115 (65.3%)

**Table 2 jcm-09-03773-t002:** Relationship between urogynecologic as well as clinical parameters and post-void residual (PVR) in 176 patients. Cut-off for significant PVR was defined as amounts >150 mL.

	PVR < 150 mL (*n* = 120)	PVR > 150 mL (*n* = 56)	*p*-Value ^1^
**BMI**			0.042
<25	31 (27.7%)	23 (43.4%)	
25–29.9	43 (38.4%)	21 (29.6%)	
≥30	38 (33.9%)	9 (17.0%)	
**Age**			0.748
<62.4	57 (47.5%)	25 (44.6%)	
≥62.4	63 (52.5%)	31 (55.4%)	
**Type of prolapse surgery**			0.089
Uterus preserving prolapse surgery	47 (39.2%)	14 (25.0%)	
Prolapse hysterectomy	73 (60.8%)	42 (75.0%)	
**POP-Q Stage**			1.000
II	33 (33.7%)	18 (33.3%)	
III and IV	65 (66.3%)	36 (66.7%)	
**ASA classification**			0.939
1	34 (28.6%)	15 (26.8%)	
2	72 (60.5%)	34 (60.7%)	
3	13 (10.9%)	7 (12.5%)	
**Colporrhaphy anterior**			0.245
Yes	91 (75.8%)	47 (83.9%)	
No	29 (24.2%)	9 (16.1%)	
**Hypertension**			1.000
No	66 (55.0%)	31 (55.4%)	
Yes	54 (45.0%)	25 (44.6%)	
**Diabetes**			0.269
No	98 (81.7%)	50 (89.3%)	
Yes	22 (18.3%)	6 (10.7%)	
**Postoperative Complication**			0.433
No	106 (88.3%)	52 (92.9%)	
Yes	14 (11.7%)	4 (7.1%)	

^1^ Chi-Square test.

**Table 3 jcm-09-03773-t003:** Multivariate regression analysis with PVR as dependent variable and clinical characteristics as independent variables.

Variable	PVR (<150 mL vs. >150 mL)		
	OR	95% CI	*p*-Value
Age	0.38	−0.01; 0.01	0.705
BMI	−2.65	−0.05; –0.01	**0.009**
OP anterior compartment	0.98	−0.1; 0.3	0.329
Duration of operation	−0.36	−0.01; 0.01	0.723
POP-Q Stage	0.34	−0.08; 0.1	0.735
Prolapse hysterectomy vs. uterus preserving prolapse surgery	−2.16	−0.38; −0.02	**0.032**

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
