# Peer review of "Uterus Preservation in Case of Vaginal Prolapse Surgery Acts as a Protector against Postoperative Urinary Retention"

_jcm, 2020, doi:10.3390/jcm9113773_

Round 1

Reviewer 1 Report

This is an interesting study in which Authors using multiple logistic regression analysis conducted in cohort of 176 operated  due to symptomatic proved independent risk factors for significant PVR after surgical prolapse repair were low BMI (p=0.009) as well as prolapse hysterectomy. Based on these findings Authors concluded that uterus preserving prolapse surgery acts as a protector against POUR and therefore uterus sparing surgery should be preferentially used  in selected prolapse cases in order to decrease possibility of POUR.

However in my opinion some pivotal issues should be explained before final approval for publication. Bulk of evidence clearly indicated that advanced prolapse may cause anatomic distortion of lower urinary tract including urethral kinking, resulted in impaired urine flow and an elevated postvoid residual [ Lukacz ES, DuHamel E, Menefee SA, et al. Elevated postvoid residual in women with pelvic floor disorders: prevalence and associated risk factors. Int Urogynecol J Pelvic Floor Dysfunct 2007;18:397–400].  According to previous studies, urinary splinting was reported by 5%to 12% of women with stage II anterior prolapse and 23% to 36% of those with stage III or IV anterior prolapse [Tan JS, Lukacz ES, Menefee SA, et al. Predictive value of prolapse symptoms: a large database study. Int Urogynecol J Pelvic Floor Dysfunct 2005;16:203–9. discussion 209]. This anatomic obstruction leads to dysfunctional voiding, which would increase the obstructive uropathy, including clinical conditions of urinary retention [Jelovsek JE, Maher C, Barber MD. Pelvic organ prolapse. Lancet 2007;369:1027–38]. It was also shown previously that the preoperative voiding study (performed with the prolapse reduced) had a sensitivity of 66%, a specificity of 46%, a positive predictive value of 12%, and a negative predictive value of 93% as  a predictor of elevated postoperative postvoid residual volumes [Postoperative resolution of urinary retention in patients with advanced pelvic organ prolapse. Fitzgerald MP, et al. Am J Obstet Gynecol 2000 Dec;183(6):1361-3; discussion 1363-4]. Therefore a preoperative voiding study (PVR) performed with the pelvic organ prolapse reduced most accurately predicted postoperative voiding function when results of the voiding study were normal.

In my opinion the data concerning preoperative PVR values in both study groups (Uterus preserving prolapse surgery, and prolapse hysterectomy) are of critical for final conclusions. Without preoperative PVR values final conclusions are in my opinion of very limited values.

Reviewer 2 Report

This is an interesting article on uterus preservation in case of vaginal prolapse surgery acts as a protector against postoperative urinary retention.

I have some comments:

Page 2

Exclusion criteria were inadequate documentation and sole correction of the posterior compartment.”

Did you include patients with previous pelvic surgery? If so, did these patients demonstrate any difference in post-op urinary retention compared to those without? If no such patients were included, you should perhaps state this.

Page 3

The cut-off value to perform clean intermittent catherization is a residual urine volume greater than 150 ml.”

How did you decide on the cut-off value? If you decided in-house, you explain how you arrived at this decision. Instead, if you used a value from literature, you should give a reference.

Page 4

Furthermore, age, type of prolapse surgery, POP-Q stage, ASA classification, anterior colporrhaphy, hypertension, diabetes and postoperative complications did not significantly differ between patients with and without PVR amounts (p>0.05) (Table 2).”

At no point do you specify what complications your patients had, which is important information relevant to the usefulness of your research. Please give a breakdown of complications and incidence.

Table 2

You used a cut-off value of 30 to evaluate the effect of BMI. This is the value for obesity, and I am curious about whether you had results also for overweight-but-not-obese, in other words with a BMI of 25-30. Did you have enough patients in this range to give results?

Table 2

Why did you set a cut-off of 62.4 for age?

Page 6

As prolapse hysterectomy tends to result in more tissue swelling and inflammation compared to sacrospinous hysteropexy, …”

You state this as given, but it would be useful to know if this is an observation from your experience, or if you have obtained it from the literature, in which case a reference would be appropriate.

Round 2

Reviewer 2 Report

You have replied exhaustively to my questions and made the modifications to the text that I requested. In my opinion the article is now ready for publication.